# SIMPLE EULERIAN-LAGRANGIAN APPROACH TO SOLVING EQUATIONS FOR SINKING PARTICULATE ORGANIC MATTER IN THE OCEAN

Vladimir Maderich<sup>1</sup>, Igor Brovchenko<sup>1</sup>, Kateryna Kovalets<sup>1</sup>, Seongbong Seo<sup>2</sup>, and Kyeong Ok Kim<sup>2</sup>

<sup>1</sup>Institute of Mathematical Machine and System Problems, Glushkov av., 42, Kyiv 03187, Ukraine

<sup>2</sup>Korea Institute of Ocean Science and Technology, Busan, Republic of Korea

**Correspondence:** Kyeong Ok Kim (kokim@kiost.ac.kr)

**Abstract.** Gravitational sinking of particulate organic matter (POM) is a key mechanism of the vertical transport of carbon in the deep ocean and its subsequent sequestration. The size spectrum of these particles is formed in the euphotic layer by primary production and various mechanisms, including food web consumption. The masses of the particles, as they descend, change under aggregation, fragmentation, and bacterial decomposition. These processes depend on the water temperature and oxygen concentration, particle sinking velocity, ages of the organic particles, ballasting and other factors. In this work, we developed a simple Eulerian-Lagrangian approach to solving equations for sinking particulate matter when the effects of the sizes and ages of the particles, temperature and oxygen concentration on their dynamics and degradation processes are considered. The model considers feedback between the degradation rate and the particle sinking velocity. We rely on known parameterizations, but our Eulerian-Lagrangian approach for solving the problem differs, which enables the algorithm to be incorporated into biogeochemical global ocean models with relative ease. Two novel analytical solutions of a system of one-dimensional Euler equations for the POM concentration and Lagrange equations for the particle mass and position were obtained for constant and age-dependent degradation rates. The feedback between the degradation rate and sinking velocity leads to significant differences in the vertical profiles of the POM concentration and sinking flux, in contrast to the solutions obtained at a constant sinking velocity, where the concentration and flux profiles of the POM are similar. The calculation results are compared with the available measurement data for the POM and POM flux for the latitude bands of 20-30°N in the Atlantic and Pacific Oceans and 50–60°S in the Southern Ocean. The dependence of the degradation rate on temperature significantly affected the profiles of the POM concentration and sinking flux by enhancing the degradation of sinking particles in the ocean's upper layer and suppressing it in the deep layer of the ocean. In all cases considered, the influence of the oxygen concentration was insignificant compared to that of the distribution of temperature with depth.

Copyright statement. TEXT

#### 1 Introduction



Gravitational sinking of particulate organic matter (POM) is a key mechanism of the vertical transport of carbon in the deep ocean (gravitational biological pump) and its subsequent sequestration (Siegel et al., 2023). The biological pump mechanism provides not only the transfer and burial of carbon but also nutrients, trace metals, and natural and artificial radionuclides through a scavenging mechanism (Roca-Martí and Puigcorbé, 2024; De Soto et al., 2018; Maderich et al., 2022). In addition to the processes of sorption and desorption, the mechanism of scavenging is controlled by the sizes of the sinking particles, their densities, the sinking velocity, and the processes of organic particle degradation (Maderich et al., 2021).

The size spectrum of the sinking particles is formed in the euphotic layer by primary production and various mechanisms, including aggregation and fragmentation under the influence of mechanical factors (Burd, 2024) and through food web consumption. The masses of the particles, as they descend in deep layers of the ocean, decrease under the influence of grazing by filter feeders and bacterial decomposition, which depends on the water temperature and oxygen concentration (Cram et al., 2018), particle falling velocity (Alcolombri et al., 2021), ages of the organic particles (Jokulsdottir and Archer, 2016; Aumont et al., 2017) and other factors, such as ballasting (Armstrong et al., 2002; Cram et al., 2018; Maerz et al., 2020). The POM degradation rate can be proportional to the particle mass (volume) (DeVries et al., 2014; Cram et al., 2018) or surface area (Omand et al., 2020; Alcolombri et al., 2021).

Many biogeochemical models assume that the settling velocity of particles is constant with depth (e.g., Aumont et al., 2015). Then, depending on the degradation rate, the vertical profiles of the POM concentration and sinking flux can be determined. At a constant degradation rate, the corresponding vertical profiles of the particle mass concentration and mass flux are exponential (Banse, 1990; Lutz et al., 2002). Assuming that the degradation rate is inversely proportional to the age of the particles (Middelburg, 1989), the vertical profiles of the particle mass concentration and mass flux can be described by a power law (Cael et al., 2021). This power law corresponds to the well-known empirical "Martin curve" (Martin et al., 1987). However, as the particle mass decreases due to degradation, the sinking velocity also decreases. This feedback, along with other factors, is taken into account in several mechanistic models (e.g., DeVries et al., 2014; Cram et al., 2018; Omand et al., 2020; Alcolombri et al., 2021); however, these models do not consider the ages of the particles.

An analytical solution to the equation for the distribution of POM by particle size was obtained by DeVries et al. (2014) for a constant degradation rate. However, as noted by DeVries et al. (2014), the values of the vertical flux of the POM mass at great depths were 1–2 orders of magnitude less than those observed. This discrepancy can be assumed to be due to the constancy of the degradation rate with depth in the model. A decrease in the rate of degradation can also be caused by a decrease in water temperature (e.g., Cram et al., 2018) or an increase in the ages of the sinking particles with depth.

In this work, we developed a simple Eulerian–Lagrangian approach for solving equations for sinking particulate matter when the effects of the sizes and ages of the particles, temperature and oxygen concentration on their dynamics and degradation processes are considered. We relied on known parameterizations (Kriest and Oshlies, 2008; DeVries et al., 2014; Cram et al., 2018), but our Eulerian–Lagrangian approach for solving the problem is different. Our approach involves solving the Euler equation for the concentration of particles of a given size and the Lagrange equations for a sinking organic particle under the

ocean models with relative ease. The remainder of the paper is organized as follows: The equations of the model for sinking particulate organic matter are presented in Sect. 2. Analytical solutions for constant and age-dependent degradation rates are obtained and compared with available data on the vertical concentration and mass flux of the POM in Sect. 3. A numerical Eulerian–Lagrangian method for the generalized model is presented in Sect. 4. The results of the simulations are discussed in Sect. 5. Our findings are summarized in Sect. 6. The equivalence of the obtained solution and the solution in (DeVries et al., 2014) for a constant rate of degradation is shown in Appendix A.

## 2 Model equations


We consider the vertical flux of organic particles caused by gravitational forces. Focusing on the development of a numerical Eulerian–Lagrangian method and finding analytical solutions, we limit ourselves to a fairly simple one-dimensional formulation of the problem away from areas of intense currents. The vertical distribution of these particles below the euphotic layer  $z_{eu}$  is governed by the flux of settling particles equilibrated by particle degradation due to bacterial decomposition. The processes of aggregation, fragmentation and ballasting are not included in the model. We limit ourselves to large-scale climatological processes that cover the water column below the euphotic layer to the bottom. We assume that the effects of time variability on the POM flux are relatively small far from this layer, and we consider the steady states of these fluxes.

The Euler particle concentration transport equation and the Lagrange equations for the individual particles are solved. The Euler equation for the POM concentration  $C_{p,d}$  [kg m<sup>-3</sup>] for particles of equivalent spherical diameter d [m] is written as

$$\frac{\partial W_{p,d}C_{p,d}}{\partial z'} + \gamma C_{p,d} = 0,\tag{1}$$

where  $W_{p,d}$  [m d<sup>-1</sup>] is the settling velocity of a particle of diameter d, z' [m] is the vertical coordinate directed downwards from the depth of the euphotic zone ( $z' = z - z_{eu}$ ), and  $\gamma$  [d<sup>-1</sup>] is the degradation rate. The boundary condition for Eq. (1) is

$$z' = 0$$
:  $C_{p,d} = C_{p,d}(0)$ , (2)

where  $C_{p,d}(0)$  is the prescribed POM concentration at the lower boundary of the euphotic layer  $z_{eu}$ .

We consider the particle dynamics in the Lagrangian coordinate system. The porosity of organic particle aggregates increases with increasing particle size (Mullin, 1966). The relationship between the organic matter mass  $m_d$  and diameter d of porous particles can be parameterized according to the particle fractal dimension

$$m_d = c_m d^{\zeta}$$
. (3)

Here,  $\zeta$  ( $\zeta \leq 3$ ) is a dimensionless scaling argument, and  $c_m$  is a prefactor coefficient (Alldredge and Gotschalk , 1988).

The Stokes-type settling velocity  $W_{p,d}$  depends on the difference between the density of water and the density of the particle, the particle shape, and the kinematic viscosity. To consider the entire ensemble of aforementioned factors that impact sinking, we approximate the sinking law by power dependence, which is widely used in particle flux models (e.g., DeVries et al., 2014):

$$W_{p,d} = \frac{\partial z'}{\partial t} = c_w d^{\eta},$$
 (4)

where t is time,  $\eta$  ( $\eta \le 2$ ) is a dimensionless scaling argument and  $c_w$  [m<sup>1- $\eta$ </sup> d<sup>-1</sup>] is a prefactor coefficient. The measurements of (McDonnell and Buesseler, 2010) show that formulations of sinking velocity as a function of only equivalent particle size can be insufficient because the shapes of the particles (e.g., faecal pellets) can significantly affect the sinking velocity. Fig. 1 from (Cael et al., 2021) also demonstrates the difficulties of describing the sinking velocities of particles of various sizes, shapes and structures with a single universal dependence. Therefore, Eq. (4) should be considered only a first approximation when describing the complex dynamics of particles.

We consider the case in which the mass of a particle that is descending with velocity  $W_{p,d}$  decreases over time t as a result of microbial degradation. This process can be described by a first-order reaction with a reaction rate of  $\gamma$  [d<sup>-1</sup>]. The corresponding equation for  $m_d$  [kg] is written as

$$\frac{\partial m_d}{\partial t} = -\gamma(\theta) m_d^{\theta}$$
. (5)

Parameter  $\theta = 1$  when the degradation rate is proportional to the particle mass, and  $\theta = 2/3$  when the degradation rate is proportional to the surface area of the particle (Omand et al., 2020).

In general, the degradation rate depends on many factors. Here, we consider only several of them: the age of the organic particle t [d], the temperature of the sea water T [ ${}^{\circ}$ C], and the concentration of oxygen  $[O_2]$  [ $\mu$ M];

$$\gamma = \gamma(t, T(z'), [O_2](z')).$$
 (6)

The parameterization used in Eq. (6) is presented in detail in Sect. 4.

#### 3 Analytical solutions

#### 3.1 Age-independent degradation rate

First, we consider the case in which the degradation rate of the particle is age independent (age-independent degradation rate (AID) model). Furthermore, we suppose that the mass loss is proportional to the mass of the particle ( $\gamma = \gamma_0$ ,  $\theta = 1$ ) and does not depend on temperature or oxygen concentration ( $\gamma_0 = \text{const}$ ). Then, the solution of Eq. (5) is

$$m_d = m_{0d} \exp\left(-\gamma_0 t\right),\tag{7}$$

where  $m_{0d} = c_m d_0^{\zeta}$  is the initial value of the particle mass for diameter  $d_0$ . Initially, the particle is placed at depth z' = 0. Combining Eq. (7) and Eq. (3) yields the change in the particle diameter over time as

$$d = d_0 \exp\left(-\frac{\gamma_0 t}{\zeta}\right)$$
. (8)

Assuming the quasiequilibrium descent of the particle in the Stokes regime, as described by Eq. (4), and taking into account that  $W_{p,d} = \partial z'/\partial t$ , we estimate the dependence of the particle depth z' on t using Eq. (8):

$$\frac{\partial z'}{\partial t} = c_w d_0^{\eta} \exp\left(-\frac{\eta \gamma_0 t}{\zeta}\right). \tag{9}$$

By integrating Eq. (9) from the initial particle depth z'=0 at t=0, we find the vertical path travelled by the particle:

$$z' = \frac{\zeta c_w d_0^{\eta}}{\eta \gamma_0} \left[ 1 - \exp\left(-\frac{\eta \gamma_0 t}{\zeta}\right) \right]. \tag{10}$$

By eliminating time from Eqs. (9) and (8) by using Eq. (10), we obtain  $W_{p,d}$  and d as functions of z':

$$W_{p,d} = H(z')c_w d_0^{\eta} (1 - \psi z'), \tag{11}$$

$$d = H(z')d_0(1 - \psi z')^{\frac{1}{\eta}},\tag{12}$$

where

$$\psi = \frac{\eta \gamma_0}{\zeta c_w d_0^{\eta}} > 0.$$
 (13)

These solutions describe a layer of finite thickness  $h_0 = \psi^{-1}$  below which there are only trivial solutions  $W_{p,d} = d = 0$ . To consider this finding, the Heaviside function is used. The Heaviside function is H(z') = 1 if  $z' \le \psi^{-1}$  and H(z') = 0 if  $z' > \psi^{-1}$ . Taking into account Eq. (11), we solve Eq. (1) with the boundary condition in Eq. (2) to obtain

$$C_{p,d} = H(z')C_{p,d}(0)\left(1 - \psi z\right)^{\frac{\zeta - \eta}{\eta}}.$$
 (14)

The solution (14) describes the vertical profile of the POM concentration for particles of diameter d under the prescribed particle size distribution  $N(d_0)$  [m<sup>-4</sup>] at z' = 0. This distribution can be approximated by the power dependence (e.g., Kostadinov et al., 2009)

$$N(d_0) = M_0 d_0^{-\epsilon},$$

where  $\epsilon$  is a power-law exponent and  $M_0$  is a constant that can be estimated from the total concentration of sinking POM at z'=0. To obtain the size distribution of  $C_{p,d}(0)$ , we use a small increment of particle size  $\Delta d_0$  under the assumption that the concentration is uniform within the interval  $\Delta d_0$ . Then, the distribution  $C_{p,d}(0)$  as a product of  $N(d_0)$ ,  $m_{0,d}$  and  $\Delta d_0$  is given by

$$C_{p,d}(0) = M_0 d_0^{-\epsilon} m_{0,d} \Delta d_0 = M_0 c_m d_0^{\zeta - \epsilon} \Delta d_0.$$
(15)

The total concentration  $C_p$  is calculated as the sum of concentrations  $C_{p,k}$  in the k-th interval of size d over the total number of  $n_d$  intervals:

$$C_p(z') = \sum_{k=0}^{n_d} C_{p,k} = M_0 c_m \sum_{k=0}^{n_d} d_{0,k}^{\zeta - \epsilon} H(z') \left(1 - \psi z'\right)^{\frac{\zeta - \eta}{\eta}} \Delta d_0, \tag{16}$$

where  $d_{0,k}=k\Delta d_0+d_0^{min}$ ,  $\Delta d_0=(d_0^{max}-d_0^{min})/n_d$ , and  $d_0^{min}$  and  $d_0^{max}$  are the minimal and maximal values, respectively, of  $d_0$ . At  $\Delta d_0 \to 0$ , the total concentration of sinking POM  $C_p$  [kg m<sup>-3</sup>] in the range from  $d_0^{min}$  to  $d_0^{max}$  can be calculated as

$$C_p(z') = M_0 c_m \int_{d_0^{min}}^{d_0^{max}} \tilde{d}_0^{\xi - \epsilon} H(z') (1 - \psi z')^{\frac{\xi - \eta}{\eta}} d\tilde{d}_0.$$
(17)

The total mass flux  $F_p$  [kg m<sup>-2</sup>d<sup>-1</sup>] can be calculated in a similar way:

$$F_p(z) = \sum_{k=0}^{n_d} C_{p,k} W_{p,k} = M_0 c_m c_w \sum_{k=0}^{n_d} d_{0,k}^{\eta + \zeta - \epsilon} H(z') \left(1 - \psi z'\right)^{\frac{\zeta}{\eta}} \Delta d_0.$$
(18)

Here,  $W_{p,k}$  is the sinking velocity in the k-th interval of size d over a total of  $n_d$  intervals.

At  $\Delta d_0 \rightarrow 0$ ,

$$F_p(z) = M_0 c_m c_w \int_{d_0^{min}}^{d_0^{max}} \tilde{d}_0^{\eta + \zeta - \epsilon} H(z') \left(1 - \psi z'\right)^{\frac{\zeta}{\eta}} d\tilde{d}_0.$$
(19)

The problem for which we obtained the solution (14) for the POM concentration  $C_{p,d}$  is similar to that solved by DeVries et al. (2014) for the particle size spectrum equation. In Appendix A, we show the equivalence of these solutions.

## 3.2 Age-dependent degradation rate

The degradation rate as a function of POM age t [d] can be described by following Middelburg (1989) as

$$\gamma(t) = \frac{\beta}{\alpha + t},\tag{20}$$

where  $\alpha$  [d] and  $\beta$  are empirical constants. We define such a model as an age-dependent degradation rate (ADD) model. The time dependencies of d and  $W_{p,d} = \partial z'/\partial t$  with parameterization of the degradation rate Eq. (20) are obtained similarly to those in Section 3.1. They are expressed as

$$d = d_0 \left(\frac{\alpha}{\alpha + t}\right)^{\beta/\zeta},\tag{21}$$

$$\frac{\partial z'}{\partial t} = c_w d^\eta \left(\frac{\alpha}{\alpha + t}\right)^{\eta \beta/\zeta}.$$
 (22)

Integrating Eq. (22) from the initial particle depth z'=0 at t=0, we find the path travelled by a sinking particle as

$$z' = c_w d_0^{\eta} \frac{\alpha \zeta}{\zeta - \eta \beta} \left[ \left( 1 + \frac{t}{\alpha} \right)^{(\zeta - \eta \beta)/\zeta} - 1 \right]. \tag{23}$$

By eliminating time from Eqs. (20), (21) and (22), we obtain depth-dependent solutions in the same way as in Eqs. (11)-(12):

$$W_{p,d}(z') = c_w d_0^{\eta} (1 + \phi z')^{-\frac{\eta \beta}{\zeta - \eta \beta}}, \tag{24}$$

$$\gamma(z') = \frac{\beta}{\alpha} (1 + \phi z')^{-\frac{\zeta}{\zeta - \eta \beta}}, \tag{25}$$

$$d(z') = d_0 (1 + \phi z')^{-\frac{\beta}{\zeta - \eta \beta}}, \tag{26}$$

where


$$\phi = \frac{\zeta - \eta \beta}{\alpha \zeta c_w d_0^{\eta}}.\tag{27}$$

By integrating Eq. (1) with the boundary condition in Eq. (2) and considering Eqs. (24) and (25), we obtain the following solution for  $C_{p,d}$ :

$$C_{p,d}(z') = C_{p,d_0} (1 + \phi z')^{\frac{(\eta - \zeta)\beta}{\zeta - \eta\beta}}$$
. (28)

The density of the distribution of the particle mass concentration at z'=0 is assumed to be approximated by a power law (15). We can obtain the total concentration of sinking POM  $C_p$  in the range of  $d_0$  from  $d_0^{min}$  to  $d_0^{max}$  as

$$C_p(z') = M_0 c_m \int_{d_0^{min}}^{d_0^{max}} \tilde{d}_0^{\xi - \epsilon} \left( 1 + \phi z' \right)^{\frac{(\eta - \xi)\beta}{\xi - \eta\beta}} d\tilde{d}_0. \tag{29}$$

The corresponding total mass flux  $F_p(z)$  is written as

$$F_p(z') = \int_{d_0^{min}}^{d_0^{max}} W_{p,d} C_{p,d} d\tilde{d}_0 = M_0 c_m c_w \int_{d_0^{min}}^{d_0^{max}} \tilde{d}_0^{\eta + \zeta - \epsilon} (1 + \phi z')^{-\frac{\zeta \beta}{\zeta - \eta \beta}} d\tilde{d}_0.$$
 (30)

#### 3.3 Comparison of analytical solutions

The obtained analytical solutions have several important properties. First, we compare these solutions with the solutions obtained under the assumption of a constant sinking velocity when

$$W_{p,d} = c_w d_0^{\eta}$$
.

The solution of Eq. (1) for a constant degradation rate  $\gamma$  corresponds to the exponential profile of the particle concentration

$$C_p(z', d_0) = C_p(0, d_0) \exp\left(-\frac{\gamma_0 z'}{c_w d_0^{\eta}}\right),$$
 (31)

whereas the time-dependent degradation rate (20) corresponds to the power-law distribution of the POM concentration

$$C_p(z', d_0) = C_p(0, d_0) \left( \frac{\alpha c_w d_0^{\eta}}{\alpha c_w d_0^{\eta} + z'} \right)^{\beta}.$$
(32)

Both of these solutions are frequently used to approximate observed particle flux profiles, e.g., (Martin et al., 1987; Lutz et al., 2002). Notably, a solution of the form (32) can alternatively be obtained under the assumption of a constant degradation rate and a linear increase in the sinking velocity (Kriest and Oshlies, 2008; Cael and Bisson, 2018).

The corresponding profiles of  $C_p$  and  $F_p$  were obtained by summing the  $n_d$  profiles in Eqs. (31) and (32). The values of  $C_p(z_{eu})$  and  $F(z_{eu})$  were calculated using Eq. (15). The model parameters  $(\eta, \zeta, \gamma_0, c_w, \epsilon, \alpha, \beta, d_0^{max}, d_0^{min}, \text{ and } n_d)$  in Table 1 were the same as in (DeVries et al., 2014) and (Aumont et al., 2017). As shown in Fig. 1, with these parameters,  $C_p$  and  $F_p$  decay much faster for the AID model than for the ADD model. Notably,  $C_p$  and  $F_p$  tend to exhibit exponential or power-law profiles only at great depths. Moreover, at a constant particle velocity, the mass-weighted sinking velocity of particles

$$\overline{W}_p(z') = \frac{F_p(z')}{C_p(z')}$$


**Table 1.** Baseline model parameters.

| Parameters        | Value/range     | Unit               | Reference                 |
|-------------------|-----------------|--------------------|---------------------------|
| $\overline{\eta}$ | 1.17            | -                  | Smayda (1970)             |
| ζ                 | 2.28            | -                  | Mullin (1966)             |
| $\gamma_0$        | 0.03            | $\mathrm{d}^{-1}$  | Kriest and Oshlies (2008) |
| $c_w$             | $2.2\cdot 10^5$ | $m^{1-\eta}d^{-1}$ | Kriest and Oshlies (2008) |
| $\epsilon$        | 4.2             | -                  | Kostadinov et al. (2009)  |
| $Q_{10}$          | 2-3             | -                  | Cram et al. (2018)        |
| $T_{ref}$         | 4               | °C                 | Cram et al. (2018)        |
| $K_O$             | 8               | $\mu$ M            | Cram et al. (2018)        |
| $\alpha$          | 30              | d                  | Aumont et al. (2017)      |
| $\beta$           | 1               | -                  | Aumont et al. (2017)      |
| $d_0^{max}$       | 2000            | $\mu\mathrm{m}$    | DeVries et al. (2014)     |
| $d_0^{min}$       | 20              | $\mu\mathrm{m}$    | DeVries et al. (2014)     |
| $n_d$             | 990             | _                  | DeVries et al. (2014)     |

increases with depth.





The presence of feedback between  $\gamma$  and  $W_{p,d}$  leads to significant changes in the  $C_p$  and  $F_p$  profiles. In the case of a constant  $\gamma_0$ , the vertical distribution of the concentration  $C_{p,d}$  for one surface fraction of POM size  $d_0$  is limited by a finite layer of thickness  $h_0 = (\zeta c_w d_0^n)(\eta \gamma_0)^{-1}$ . The particles in this layer sink at a linearly decreasing velocity. The masses of the particles also decrease with depth until a depth at which they are completely remineralized is reached. The size distribution for a single particle at depth z' is  $N(d,z') = C_{p,d}m_d^{-1} \sim (1-\psi z')^{-1}$ . At  $z' \to h_0$ ,  $N(d,z') \to \infty$  as  $m_d \to 0$ . The finite thickness of the layer of sinking particles with the parameters given in Table 1 varies from 45.4 m at  $d_0 = 20~\mu m$  to 9937 m at  $d_0 = 2000~\mu m$ . Notably, the solution to the problem in a different formulation (Omand et al., 2020) has the same qualitative character. However, the total POM concentration and total POM flux decay asymptotically approaching exponential profiles , in contrast to the profiles (14) and (11) for one class of particle sizes  $d_0$  on z'=0. The total concentration and flux profiles, normalized to values at the base of the euphotic layer, are shown in Fig. 1, where the  $C_p$  and  $F_p$  profiles were obtained via the summation of the  $n_d$  profiles in Eqs. (16) and (18). The baseline parameters for the calculation are presented in Table 1. These parameter values match those used by DeVries et al. (2014). Therefore, the curves in Fig. 1 also coincide with the corresponding curves in Fig. 1c from (DeVries et al., 2014), which were calculated using an equivalent formulation of the same problem, as shown in the Appendix.

In contrast to the AID model solution (16), the POM concentration profile (28) decays asymptotically with depth at  $\zeta > \eta$  and  $\zeta > \eta\beta$  for the ADD model (20). These conditions are met for the parameters listed in Table 1. The rate of degradation  $\gamma$  also decays with depth. Unlike the models (Kriest and Oshlies, 2008; Cael and Bisson, 2018) that use the same "Martin curve" power-law dependence (32) for the concentration and mass flux of POM with the exponent  $\beta$ , the exponent in the obtained

Figure 1. Normalized total POM concentration  $C_p$  (a), total POM flux  $F_p$  (b), and weighted vertical velocities of the particles (c) for the AID (blue lines) and ADD (red lines) models calculated from the analytical and numerical solutions. The dashed lines correspond to the solutions of (1) at constant  $W_p(d_0)$ , whereas the solid line corresponds to the solution of the problem at variable  $W_{p,d}$ . The small circles correspond to the numerical solutions obtained via the AID and ADD models.

solution (28) depends not only on  $\beta$  but also on the parameters that characterize the sinking velocity ( $\eta$ ) and the particle mass fractal dimension ( $\zeta$ ).

The sensitivity of the AID model parameters was considered by DeVries et al. (2014). They reported that four parameters  $(\eta, \zeta, \gamma_0, \text{ and } \epsilon)$  control the flux profile and that the most significant factor is the slope of the particle distribution  $\epsilon$  on z' = 0, which has the greatest influence on the depth distribution of the particles. In Fig. 2, the variables are presented in logarithmic coordinates. Only the vertical distribution of  $C_p$  is close to the power distribution with an exponent of approximately 1, whereas the distribution with depth of  $F_p$  significantly deviates from power law (Martin's law). The sensitivity of the concentration and flux profiles to the values of parameters  $\alpha$  and  $\beta$  is examined in Figure 2. An increase in  $\alpha$  leads to a deepening of the concentration and particle flux profiles, whereas an increase in  $\beta$  leads to a shallowing of these profiles.

The relative maximal absolute errors [%] of the calculated AIDR and ADDR solutions for  $C_p$  and  $F_p$  are presented in Table S1. We compare the solutions at spectral resolutions  $n_d = 100$  and  $n_d = 10$  with the baseline calculation at  $n_d = 990$ . These estimates demonstrate the necessity of fine resolution of the spectre of particles at the lower boundary of the euphotic zone for obtaining accurate profiles of the POM concentration and sinking flux. In this case, the particle concentration profile is more sensitive to the spectral resolution than the sinking flux profile is.

#### 4 Numerical model



#### 4.1 Numerical algorithm

The model discussed in the previous section is based on several simplifying assumptions that make obtaining analytical solutions to the system of equations possible. However, when we expand the model to include new important factors in the processes of sinking and remineralization of POM, analytical solutions to the problem can no longer be obtained. Therefore, a new numerical Eulerian–Lagrangian approach for solving this problem was developed.

Figure 2. Sensitivity of the normalised total particle concentration  $C_p$  (a) and total particle flux  $F_p$  (b) to parameters  $\alpha$  and  $\beta$ .

Here, we consider the case in which the degradation rate depends on the age of the organic particle (ADD model), the temperature of the sea water T and the concentration of oxygen  $[O_2]$ :

$$\gamma = \gamma(t, T(z'), [O_2](z')) = \left(\frac{\beta}{\alpha + t}\right) \left(Q_{10}^{\frac{T - T_{ref}}{10}}\right) \left(\frac{[O_2]}{K_O + [O_2]}\right),\tag{33}$$

where  $Q_{10}$  is the temperature coefficient,  $T_{ref}$  is a reference temperature, and  $K_O$  [ $\mu$ M] is an oxygen dependence parameter (Cram et al., 2018). When  $\gamma$  does not depend on age (AID model), then

$$\gamma = \gamma_0 \left( Q_{10}^{\frac{T - T_{ref}}{10}} \right) \left( \frac{[O_2]}{K_O + [O_2]} \right). \tag{34}$$

The system of Lagrange equations for particle depth and size derived from Eqs. (4)-(5) is as follows:

$$\frac{\partial d}{\partial t} = -\frac{\gamma(t, T(z'), [O_2](z'))}{\zeta} d, \tag{35}$$

$$\frac{\partial z'}{\partial t} = c_w d^{\eta}. \tag{36}$$

The initial conditions are that at t = 0: z' = 0 and  $d = d_{0,i}$ .

The procedure for determining the profiles of  $C_p(z')$  and  $F_p(z')$  is presented in Fig. 3. It includes 11 steps.

Step 1 The model parameters and temperature and oxygen concentration profiles are read from the input files.

Step 2 A regular Eulerian grid  $\bar{z}'$  is established from 0 to the ocean depth D with  $n_z$  equal intervals  $\Delta z$  with levels  $\bar{z}' = j \cdot \Delta z$ ,

where  $j = (0, n_z)$ . The particle size spectrum at the lower boundary of the euphotic layer is divided into  $n_d$  equal intervals of size  $\Delta d$  in the range from  $d_{min}$  to  $d_{max}$ . For every particle size  $d_{0,k}$ ,  $k = (0, n_d)$ .

Figure 3. Eulerian-Lagrangian method flow chart for equations of sinking particulate organic matter.


Step 3 Steps 4–9 are performed for every  $d_{0,k}$ , where  $k=(0,n_d)$ . Then, Step 10 is performed.

Step 4 The initial conditions are set for the Lagrangian particle depth  $\tilde{z_k}'(t_i)$  and size  $d = d_{0,k}$  equations at  $t_i = 0$ .

Step 5 If the Lagrangian particle depth  $\tilde{z_k}'(t_i)$  is equal to or greater than the ocean depth D or the particle diameter at this depth level  $d = d_{0,k}$  is equal to or less than 1% of the minimum diameter  $d_{min}$ , Step 8 is performed; otherwise, Step 6 is performed.

Step 6 The timescale is divided into intervals  $\Delta t_i$ ,  $i=(0,n_t)$  over which Eqs. (35)-(36) are integrated. To align the resulting  $\tilde{z}_k'(t_i)$  and regular depth grid  $\bar{z}'$ , the i-th timestep duration is calculated as  $\Delta t_{i,k} = \Delta z/(c_w d_{i,k}^\eta)$ .

Step 7 The Lagrangian formulation with respect to time t is used to solve the system of equations (35)-(36) via the Runge–Kutta method of the 4th order. Cubic spline interpolation is used to calculate the temperature and oxygen concentration at  $\tilde{z_k}'(t_i)$ .

Step 8 The  $w_p(\tilde{z_k}'(t_{i+1}))$  and  $\gamma(\tilde{z_k}'(t_{i+1}))$  profiles on the Lagrangian grid are interpolated via a cubic spline over the Eulerian grid  $\bar{z}'$ .

Step 9  $C_{p,d}(\bar{z}')$  is calculated by solving the Euler equation (1) via the Runge–Kutta method over the regular grid  $\bar{z}'$ . Then, Step 3 is performed.

Step 10 The total POM concentration  $C_p(\bar{z}')$  and POM flux  $F_p(\bar{z}')$  are obtained via numerical integration of  $S_p(p,d)(\bar{z}')$  and  $w_p(\bar{z}')$  by using the composite Simpson's 1/3 rule.

Step 11 The model outputs the total POM concentration  $C_p(\bar{z}')$  and POM flux  $F_p(\bar{z}')$ .

The code for the proposed algorithm, along with the data used in this study, is archived on Zenodo (Kovalets et al., 2025a, b).

#### 4.2 Numerical model setup






Simulations were carried out for a water column with a depth of D=5000 m and  $\Delta z=1$  m. We calculated the vertical profiles of the POM concentration  $C_p$  and flux  $F_p$  using AID and ADD models for the degradation rate. The remaining model parameters, with the exception of  $\eta$ , were adopted from Table 1. The profiles of  $C_p$  and  $F_p$  were calculated via the above algorithm with  $\eta=1.17$  for comparison with the analytical solutions with the AID and ADD parameters from Table 1. As shown in Fig. 1, the numerical and analytical profiles coincide.

The calculation results were compared with the available measurement data for  $C_p$  and  $F_p$  for the latitude bands of 20– $30^\circ N$  in the Atlantic and Pacific Oceans and 50– $60^\circ S$  in the Southern Ocean. These calculations aimed to assess the relative effects of the vertical distributions of temperature and oxygen in the Atlantic, Pacific and Southern Oceans on the profiles of  $C_p$  and  $F_p$ . For the Atlantic Ocean, the  $C_p$  and  $F_p$  data are compiled in (Aumont et al., 2017) and (Lutz et al., 2002). For the Pacific Ocean, these values are presented in (Martin et al., 1987) and (Druffel et al., 1992). The Southern Ocean data for the Pacific and Atlantic sectors are presented in (Aumont et al., 2017) and (Lutz et al., 2002). The calculations required for averaging over the region and time profiles of T and  $[O_2]$  were performed with the measurement data from (Boyer et al., 2018). These averaged profiles are shown in Fig. S1 in the Supplement. Notably, there is great uncertainty not only in the choice of model parameter values but also in the parameterization of the processes. This is explained by both an insufficient understanding of the physical and biogeochemical processes and the lack of a sufficient number of measurements in the deep layers of the ocean. In particular, the observation results (Cael et al., 2021) show large deviations in the parameters of the sinking velocity–particle size relationship (4). In recent models, the parameter  $\eta$  has varied from 0.26 (Alcolombri et al., 2021) to 2 (Omand et al., 2020). Therefore, in the simulations, we compared the effects of  $\eta$  on the  $C_p$  and  $F_p$  profiles for two values:  $\eta = 1.17$  (Smayda, 1970) and  $\eta = 0.63$  (Cael et al. , 2021).

#### 5 Modelling results

#### 5.1 Comparison of simulations with measurements

Figures 4–6 show the profiles of  $C_p$  and  $F_p$  normalized to  $C_p(z_{eu})$  and  $F_p(z_{eu})$ . They were calculated using the numerical algorithm described in Sect. 4.1. These profiles are compared with normalized measurements in the subtropical zones of the Atlantic (Fig. 4) and Pacific (Fig. 5) Oceans and in the Atlantic and Pacific sectors of the Southern Ocean (Fig. 6) to consider the effects of temperature and oxygen concentration on POM. When the modelling results are compared with the measurement data, the significant scatter of the measurement data presented in Figs. 4–6 must be noted. This scatter is due both to the

Figure 4. Normalized total POM concentration  $C_p$  (a–c) and total POM flux  $F_p$  (d–f) versus measurement data in the Atlantic Ocean at 20-30°N (Aumont et al., 2017; Lutz et al., 2002). Three columns of panels correspond to the model without dependency of temperature and oxygen (panels a and d), additional temperature dependence (panels b and e), and both additional dependencies (panels c and f).

difficulties of measuring the concentration and flux of particles and to regional differences in the influx of particles and in the surrounding ocean.

The  $C_p$  and  $F_p$  profiles in Figures 4–6 were obtained for three variants of the degradation model. In the first variant (plots a and d),  $C_p$  and  $F_p$  do not depend on the temperature or oxygen concentration. In the second variant (plots b and e), they do not depend on the oxygen concentration, and in the third variant (c and f), they depend on the temperature and oxygen concentration. The first variant is described by analytical solutions for the AID and ADD models. The features of these solutions are discussed in Section 3.3. The profiles of  $C_p$  and  $F_p$  are sensitive to the value of  $\eta$ . The solutions with  $\eta = 0.63$  decay more slowly than those with  $\eta = 1.17$  do, as shown by the analytic solutions in Figs. 4a, 4d, 5a, 5d, 6a, and 6d.



The use of the AID model led to a more rapid decay of  $C_p$  with depth than was observed in all the ocean profiles. Moreover, the application of the ADD model resulted in smoother profiles in all oceans; however, the AID and ADD profiles are qualitatively close. As shown in Figs. 4b, 4e, 5b, 5e, 6b, and 6e, the dependence of the degradation rate on temperature significantly affected the  $C_p$  and  $F_p$  profiles; namely, it enhanced the degradation of sinking particles in the upper layers of the ocean and

Figure 5. Normalized total POM concentration  $C_p$  (a–c) and total POM flux  $F_p$  (d–f) versus measurement data in the Pacific Ocean at 20-30°N (Martin et al., 1987; Druffel et al., 1992). Three columns of panels correspond to the model without dependency of temperature and oxygen (panels a and d), additional temperature dependence (panels b and e), and both additional dependencies (panels c and f).

suppressed it in the deep layers of the ocean. The influence of the oxygen concentration in all the cases considered (Figs. 4c, 4f, 5c, 5f, 6c, and 6f) was less significant than that of the distribution of temperature with depth. Overall, including temperature and concentration dependence in the degradation rate relationship improved the agreement with ocean measurements. The normalized mean bias errors (MBEs) when considering the dependence of the degradation rate on temperature and oxygen concentration (third variant) decreased from 9% to -3% compared to those of the first variant, when this dependence was not considered. For the third variant, the root mean square deviation (RMSD) decreased by half compared with that of the first variant.


Notably, both the AID and ADD models somewhat underestimated  $F_p$  when the dependence on temperature was considered. As shown in Figs. 4–6, the use of the AID model led to a more rapid decay of  $C_p$  with depth than was observed in all ocean profiles. Moreover, the decay of  $F_p$  with depth occurred more slowly in most of the measured profiles. The use of the ADD model (Figs. 4–6) resulted in smoother profiles; however, qualitatively, the AID and ADD profiles are similar. Notably, profiles  $C_p$  and  $F_p$  in Fig. 3c, 3f and 4c, 4f are quite close despite the differences between the temperature and oxygen concentration

**Figure 6.** Normalized total POM concentration  $C_p$  (a–c) and total POM flux  $F_p$  (d–f) versus measurement data in the Southern Ocean at 50-60°N (Aumont et al., 2017; Lutz et al., 2002). Three columns of panels correspond to the model without dependency of temperature and oxygen (panels a and d), additional temperature dependence (panels b and e), and both additional dependencies (panels c and f).

profiles in the 20-30°N band of the Atlantic and Pacific Oceans (Fig. S1a-S1b). These profiles in the colder, oxygen-saturated waters of the Southern Ocean (Fig. S1c) attenuate more slowly with depth.

# 5.2 Sensitivity study

As shown in Figs. 4–6, the model output was sensitive to parameters with high uncertainty, such as  $\eta$ . Therefore, a sensitivity study was carried out for the model parameters in Table S1. We used the one-at-a-time method to quantify the effect of variation in a given parameter on the model output while all other parameters were kept at their initial values (Hamby, 1994; Lenhart et al., 2002; Soares and Calijuri, 2021). The effects of variations in these parameters were estimated for the particle transfer efficiency (TE).  $TE_{1000}$  is defined as the ratio of the POM flux at the lower level of the euphotic layer  $z_{eu}$  to the flux at the lower boundary of the mesopelagic layer z=1000, and  $TE_{5000}$  is defined as the ratio of the POM flux at the lower level of the euphotic layer to the flux in the bottom layer at z=5000m. The ranges for the parameters were defined for a constant ratio t=1000 ratio t=10000 ratio t=100000 ratio t=100000 ratio t=100000 ratio t=100000 ratio t=100000 ratio t=1000000 ratio t=1000000 ratio t=10000000 r

1/r, whereas the maximum value  $p_{max}$  was set to be proportional to the reference value  $p_{ref}$  with a ratio value of r. For the parameters in Table S1, the value of r was chosen to be the same (r = 1.25), which satisfies the ranges of all the parameters.

The model output sensitivity was estimated using a sensitivity index (SI) defined as

$$SI = \frac{TE(p_{max}) - TE(p_{min})}{TE(p_{ref})},$$
(37)

where  $TE(p_{max})$ ,  $TE(p_{min})$  and  $TE(p_{ref})$  are the simulation results for the maximal  $p_{max}$ , minimal  $p_{min}$ , and reference  $p_{ref}$  parameter values, respectively. Calculations of SI were carried out for the Pacific Ocean for the AID and DDR models with the reference, maximal and minimal values of the parameters from Table S2.

The sensitivity index  $SI(TE_{1000})$  is shown in Fig. S2 in the Supplement for the parameters of the AID model. As shown in the figure,  $SI(TE_{1000})$  was most sensitive to the exponent  $\zeta$  in the power law dependence of the particle mass on the particle size (3) and to the exponent  $\epsilon$  in the power law dependence of the particle size distribution at the lower boundary of the euphotic layer (15). The sign of the index indicates whether the model reacted codirectionally to the input parameter change, i.e., whether the parameter increase/decrease corresponded to an increase/decrease in the model output parameter. The nature of the dependence of  $TE_{1000}$  on  $\zeta$  and  $\epsilon$  was different. An increase in  $\zeta$  resulted in an increase in  $TE_{1000}$ , i.e., an increase in the mass of a particle increased the transfer efficiency. Moreover, an increase in  $\epsilon$  resulted in a decrease in  $\epsilon$  in  $\epsilon$  increase in the slope of the spectral particle size distribution led to a decrease in the transfer efficiency. The dependence of  $\epsilon$  increase in the slope of the spectral particle size distribution led to a decrease in the transfer efficiency. The dependence of  $\epsilon$  increase in the slope of the spectral particle size distribution led to a decrease in the transfer efficiency. The sensitivity index  $\epsilon$  in Fig. S3. As shown in the figure, it is qualitatively similar to that in Fig. S2. Four parameters ( $\epsilon$ 0,  $\epsilon$ 1,  $\epsilon$ 2, and  $\epsilon$ 3 showed strong sensitivity.

The sensitivity index  $SI(TE_{1000})$  values for the parameters of the ADD model are shown in Fig. S4. Similar to the results in Fig. S2,  $TE_{1000}$  was most sensitive to  $\zeta$  and  $\epsilon$ ; however, the amplitudes of  $SI(TE_{1000})$  were less than those for AID model. The sensitivity of the ADD model parameters ( $\alpha$  and  $\beta$ ) was moderate. The sensitivity index  $SI(TE_{5000})$  values for the parameters of the ADD model are shown in Fig. S5. Similar to  $SI(TE_{1000})$  for this model, the magnitudes of the  $SI(TE_{5000})$  values were greater than the magnitudes of the  $Si(TE_{5000})$  values. Additional details on the sensitivity study are presented in the Supplement.

#### 6 Discussion and conclusions





In this work, we considered a simple Eulerian–Lagrangian approach for solving equations that describe the gravitational sinking of organic particles under the effects of the sizes and ages of the particles, temperature and oxygen concentration on their dynamics and degradation processes. In contrast to other approaches, our approach does not use particle spectrum equations (e.g., DeVries et al., 2014) explicitly or power-law particle size distribution assumptions (e.g., Kriest and Evans, 1999; Maerz et al., 2020). Unlike (Omand et al., 2020), we do not assume *a priori* the constancy of the particle flux in depth in steady state. Instead, solutions are found for the Euler equation for the concentration of particles of a given size and the Lagrange equations for a sinking organic particle under the influence of microbiological degradation. In the stationary case, the problem is reduced

to solving a system of ordinary differential equations of the first order, in contrast to (DeVries et al., 2014), where the solution of the hyperbolic equation of the first order for the particle distribution is found. In addition, the total concentration and flux of the POM are found by summation over the particle distribution at z'=0, whereas in (DeVries et al., 2014) the summation is carried out over all depths. Our approach makes the particle transport model compatible with large-scale biogeochemical models and provides an opportunity to solve the non-stationary problem in the future using Eq. (1) complemented by the time derivative of  $C_{p,d}$  and necessary parameterizations of the POM sinking processes.







Novel analytical solutions of the system of the one-dimensional Eulerian equation for the POM concentration and Lagrangian equations for the particle mass and depth were obtained for constant and age-dependent degradation rates. The feedback between the degradation rate and sinking velocity results in significant changes in the POM concentration and flux profiles. In the case of a constant  $\gamma_0$  (AID model), the vertical distribution of the concentration  $C_{p,d}$  for a single fraction of the POM size  $d_0$  at  $z_{eu}$  is limited by a finite layer, unlike the exponential profile of the particle concentration that corresponds to a constant sinking velocity. Particles in such a finite layer sink at a linearly decreasing velocity. Moreover, the distributions of the total particle concentration  $C_p$  and flux  $F_p$  approach exponential trends with depth for increasing  $d_0$  fractions.

In contrast to those for the AID model, the vertical distributions of the concentration and vertical velocity decay asymptotically with depth for the ADD model. The rates of degradation of the Eulerian variables decay with depth; however, the corresponding exponent depends not only on the parameter  $\beta$ , as in the models with constant sinking velocity (Cael et al., 2021), but also on the parameters that characterize the vertical velocity  $\eta$  and porosity  $\zeta$  of the particles. With the baseline parameters, the vertical distribution of  $C_p$  is close to the power distribution with an exponent of approximately 1, whereas the distribution with the depth of the total particle flow  $F_p$  deviates significantly from the power law dependence ("Martin's law"). Direct comparison with other models is difficult owing to differences in the parameterizations of processes, with the exception of the model (DeVries et al., 2014) for which the solutions of the equations for the particle spectrum and concentration are established (Appendix A).

A new Eulerian–Lagrangian numerical approach for solving the problem in general cases was presented. The algorithm includes time steps for Lagrangian variables (sinking velocity and particle mass) and Eulerian depth steps for the concentration of particles of size d. This enables the inclusion of different parameterizations of interacting degradation and sinking processes (e.g., DeVries et al., 2014; Cram et al., 2018; Omand et al., 2020; Alcolombri et al., 2021). However, in this study, we limited ourselves to the case where the degradation rate depends on the age of the organic particle, the temperature of the sea water and the concentration of oxygen. Notably, the developed numerical algorithm is suitable for arbitrary dependencies of mass and sinking velocity on the particle diameter. The proposed numerical method was tested on the obtained analytical solutions.

The calculation results were compared with the available measurement data for the POM and POM fluxes for the latitude bands of 20–30°N in the Atlantic and Pacific Oceans and 50–60°S in the Southern Ocean. The dependence of the degradation rate on temperature affects the profiles of the total particle concentration and flux significantly; it enhances the degradation of sinking particles in the upper layers of the ocean and suppresses it in the deep layers of the ocean. Overall, including temperature and concentration dependence in the degradation rate relationship improves the agreement with ocean measurements. In particular, the normalized MBEs when considering the dependence of the degradation rate on temperature and oxygen concen-

tration were reduced from 9% to -3% compared with cases in which this dependence was not taken into account. Similarly, on average, the RMSD decreased by half when temperature stratification was considered.





The discrepancies between the model predictions and observations were caused by incomplete descriptions of processes and uncertainties in model parameters, as well as variability in the measured POM concentration and flux profiles owing to vertical and horizontal variability in the ocean fields. We used the one-at-a-time method to quantify the effect of the variation of one parameter from the set  $(\gamma_0, \eta, \zeta, \epsilon, T_{ref}, Q_{10}, K_0, \alpha, \beta)$  on the model output, with all other parameters kept at their initial values. The effects of variations in these parameters on the particle transfer efficiency TE were estimated as the ratio of the POM flux at  $z_{eu}$  to the value at a depth of 1000 m or 5000 m. The model output sensitivity was estimated via the sensitivity index SI (37). Calculations for the Pacific Ocean revealed that  $TE_{1000}$  and  $TE_{5000}$  are most sensitive to the parameters  $\zeta$  and  $\epsilon$ , respectively, for both models. Therefore, these parameters should be primarily calibrated and optimized. Therefore, it was important to assess the sensitivity of the calculations to the values of the model parameters.

Notably, to obtain analytical solutions and demonstrate the numerical Eulerian–Lagrangian approach, significant simplifications were made in the description of the particle dynamics. In particular, the particle sinking velocity was described in the Stokes approximation. The aggregation and fragmentation of particles, mineral ballasting, ocean density stratification, and temporal changes in particle flows were not considered. While some simplifications can be eliminated by using a numerical approach, others require significant generalization. This applies particularly to the description of particle ballasting mechanisms. On the one hand, ballast affects the sinking of particles, but on the other hand, ballast minerals can protect organic matter from degradation ((Cram et al., 2018)). The processes of fragmentation and consumption of sinking particles, which are important in the upper mesopelagic layer, are poorly understood (Burd , 2024). Comparison of calculation results for different parameter values (e.g.  $\eta$ ) did not reveal the advantage of one parameter value for both  $C_p$  and  $F_p$ , which may be due to the incompleteness of the description of the processes of the simplified model used. Therefore, for the effective application of the proposed approach in biogeochemical models, a parameterization of the main process controls of the biological carbon pump mechanism based on data from natural and laboratory measurements is necessary.

*Code and data availability.* The exact version of the model that was used to produce the results presented in this paper is archived on Zenodo (Kovalets et al., 2025a),

and the input data that were used to run the model and generate the plots for all the simulations presented in this paper were obtained from (Kovalets et al., 2025b).

#### Appendix A: Derivation of the spectral solution for the size distribution (DeVries et al., 2014)

Here, we show that the analytical solution (14) for  $C_{p,d}$  is equivalent to the solution (8) from (DeVries et al., 2014) of the spectral equation for the particle size distribution. To find the particle size distribution at z', we first rearrange Eq. (12) to

obtain the relationship between  $d_0$  and d at depth z':

$$d_0 = d(1 + \psi z')^{1/\eta}. \tag{A1}$$

The size distribution N(d,z) [m<sup>-4</sup>] is related to  $C_{p,d}$  and  $m_d$  as

$$C_{p,d} = Nm_d \Delta d, \tag{A2}$$

where  $\Delta d$  is a small increment. Combining Eqs. (14), (4), (12), and (A2) yields

$$N(d,z') = \frac{C_{p,d}}{m_d \Delta d} = \frac{\Delta d_0}{\Delta d} M_0 d_0^{-\epsilon} \left( 1 + \frac{\eta \gamma_0}{\zeta c_w d^{\eta}} z' \right)^{\frac{\eta - \epsilon}{\eta}}. \tag{A3}$$

At the limit of  $\Delta d \rightarrow 0$ , we obtain

$$\lim_{\Delta d \to 0} \frac{\Delta d_0}{\Delta d} = \frac{d}{dd} d_0 = \left( 1 + \frac{\eta \gamma_0}{\zeta c_w d^{\eta}} z' \right)^{\frac{1-\eta}{\eta}}.$$
(A4)

Then, Eq.(A3) can be written as

$$N(d,z') = M_0 d^{-\epsilon} \left( 1 + \frac{\eta \gamma_0}{\zeta c_w d^{\eta}} z' \right)^{\frac{1-\epsilon}{\eta}}.$$
 (A5)

This solution for N coincides with that obtained by DeVries et al. (2014).

Author contributions. VM-conceptualization; VM and IB-methodology; KK and KOK-software; SS-visualization; KOK, KK, and SS-investigation; VM, IB, and KK-writing (original draft); KOK and SS-writing (review and editing); and VM-supervision. All authors contributed to the interpretation of the findings and the writing of the paper.

Competing interests. The corresponding author declares that none of the authors have any competing interests.

Acknowledgements. The research is supported by the National Research Foundation of Ukraine (project no. 2023.03/0021), the Korea Institute of Marine Science and Technology Promotion (KIMST) funded by the Ministry of Oceans and Fisheries (RS-2023-00256141) and , the European Union's Horizon 2020 research and innovation framework program (PolarRES, Grant Agreement 101003590).

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
