# Peer review of "SIMPLE EULERIAN-LAGRANGIAN APPROACH TO SOLVING EQUATIONS FOR SINKING PARTICULATE ORGANIC MATTER IN THE OCEAN"

_EGUsphere, 2025_

## Author Comment (AC4)

**Response to Reviewer #3**

*This paper presents a combined Eulerian-Lagrangian formulation of the organic particles descent while taking into account their size and age among other things, resulting in a non-trivial relation among them, unlike suggested by previous studies. This is a well-written article with strong positioning regarding motivation, contribution and place among the previous literature. My detailed major and minor comments, mainly refraining to comment on the formulations, are organized below in the order of appearance in the manuscript.*

**Answer.** Thank you very much for the very thoughtful comments on our paper. We have followed your suggestions and revised the manuscript accordingly. Please find our responses below.

*While the contribution is clear around line 45, I would recommend highlighting the novelty by explaining the unique contribution in the context of limitations of previous attempts in the literature.*

**Answer.** The novelty of our study is the development of the Euler-Lagrangian approach and the application of the corresponding numerical algorithm to solve the problem. We have added explanatory text.

L. 45 "We solved the Eulerian equation for the concentration of particles of a given size and the Lagrangian equations for a sinking organic particle under the influence of microbiological degradation. It allows incorporation of the proposed algorithm into biogeochemical global ocean models with relative ease."

L. 203 "Unlike the models (Kriest and Oshlies, 2008; Cael and Bisson, 2018) that use the same "Martin curve" power-law dependence (32) for the concentration and mass flux of POM with the exponent $\beta$, the exponent in the obtained solution (28) depends not only on $\beta$ but also on the parameters that characterize the sinking velocity ($\eta$) and the particle mass fractal dimension ($\zeta$)."

L. 345 "In this work, we considered a simple Eulerian–Lagrangian approach for solving equations that describe the gravitational sinking of organic particles under the effects of the sizes and ages of the particles, temperature and oxygen concentration on their dynamics and degradation processes. In contrast to other approaches, our approach does not use particle spectrum equations (e.g., DeVries et al., 2014) explicitly or power-law particle size distribution assumptions (e.g., Kriest and Evans , 1999; Maerz et al., 2020). Unlike (Omand et al., 2020), we do not assume *a priori* the constancy of the particle flux in depth in the steady state problem. Instead, solutions are found for the Euler equation for the concentration of particles of a given size and the Lagrange equations for a sinking organic particle under the influence of microbiological degradation. In the stationary case, the problem is reduced to solving a system of ordinary differential equations of the first order, in contrast to (DeVries et al., 2014), where the solution of the hyperbolic equation of the first order for the particle distribution is found. In addition, the total concentration and flux of the POM are found by summation over the particle distribution at $z' = 0$, whereas in (DeVries et al., 2014) the summation is carried out over all depths. Our approach makes the particle transport model compatible with large-scale biogeochemical models and provides an opportunity to solve the non-stationary problem in the future using equation (1) for different parameterizations of the POM sinking processes."

L. 358 "Novel analytical solutions of the system of the one-dimensional Eulerian equation for the POM concentration and Lagrangian equations for the particle mass and depth were obtained for constant and age-dependent degradation rates…"

L. 374 "A new Eulerian–Lagrangian numerical approach for solving the problem in general cases was presented. The algorithm includes time steps for Lagrangian variables (sinking velocity and particle mass) and Eulerian depth steps for the concentration of particles of size $d$. This enables the inclusion of different parameterizations of interacting degradation and sinking processes (e.g., DeVries et al., 2014; Cram et al., 2018; Omand et al., 2020; Alcolombri et al., 2021). However, in this study, we limited ourselves to the case where the degradation rate depends on the age of the organic particle, the temperature of the sea water and the concentration of oxygen. Notably, the developed numerical algorithm is suitable for arbitrary dependencies on the particle diameter. The proposed numerical method was tested on the obtained analytical solutions."

*One major comment I have is that the modeling results section needs to be more specific in terms of presenting the results. The authors are showing and referring to the results but not describing and discussing them.*

**Answer.** Thank you for the comment. We extended the description of results and added a discussion in Sect. 6 (L. 318).

L. 307  "Notably, profiles $C_p$ and $F_p$ in Fig. 3c, 3f and 4c, 4f are quite close despite the differences between the temperature and oxygen concentration profiles in the 20-30 ∘N band of the Atlantic and Pacific Oceans (Fig. S1a-S1b). These profiles in the colder, oxygen-saturated waters of the Southern Ocean (Fig. S1c) attenuate more slowly with depth."

*Same goes for the validations -- I found it hard to follow which parts agree with the measurements more than others. Both of the aspects need to be addressed in the results section for the paper to be strong.*

**Answer.** We added estimates of the model accuracy for different spectral resolutions of the particle sizes $n_d$ :

L. 216 "The relative maximal absolute errors [%] of the calculated AIDR and ADDR solutions for $C_p$ and $F_p$ are presented in Table S1. We compare the solutions at spectral resolutions $n_d = 100$ and $n_d = 10$ with the baseline calculation at $n_d = 990$. These estimates demonstrate the necessity of fine resolution of the spectre of particles at the lower boundary of the euphotic zone for obtaining accurate profiles of the POM concentration and sinking flux. In this case, the particle concentration profile is more sensitive to the spectral resolution than the sinking flux profile is."

When comparing simulation results and measurement data, it is important to keep in mind that (L. 389) "discrepancies between the model predictions and observations were caused by incomplete descriptions of processes and uncertainties in model parameters, as well as variability in the measured POM concentration and flux profiles owing to vertical and horizontal variability in the ocean fields."

L. 283 "When the modelling results are compared with the measurement data, the significant scatter of the measurement data presented in Figs. 4--6 must be noted. This scatter is due both

to the difficulties of measuring the concentration and flux of particles and to regional differences in the influx of particles and in the surrounding ocean."

*When saying Fig 4-6, as in line 270, I'd recommend being specific when referring to figures and panels to show results.*

**Answer.** We referred figures accordingly to your suggestion.

L. 281 "These profiles are compared with normalized measurements in the subtropical zones of the Atlantic (Fig. 4) and Pacific (Fig. 5) Oceans and in the Atlantic and Pacific sectors of the Southern Ocean (Fig. 6) to consider the effects of temperature and oxygen concentration on POM."

L. The solutions with $\eta = 0.63$ decay more slowly than those obtained for $\eta = 1.17$ as follows also from analytic solutions in Figs. 4a, 4d, 5a, 5d, 6a, 6d.

L. 296 "As shown in Figs. 4b, 4e, 5b, 5e, 6b, and 6e, the dependence of the degradation rate on temperature significantly affected the $C_p$ and $F_p$ profiles; namely, it enhanced the degradation of sinking particles in the upper layers of the ocean and suppressed it in the deep layers of the ocean."

*"As follow" is a typo in line 272*

**Answer.** Thank you. We corrected this typo.

*Another major comment is that the paper is missing the significance in terms of closing the loop and relating back the findings back to the implementation in the earth system/biogeochemical models as the authors had described in the introduction section.*

**Answer.** We reworked and added text accordingly:

L. 355 "Our approach makes the particle transport model compatible with large-scale biogeochemical models and provides an opportunity to solve the non-stationary problem in the future using equation (1) for different parameterizations of the POM sinking processes."

L. 398 "Notably, to obtain analytical solutions and demonstrate the numerical Eulerian--Lagrangian approach, significant simplifications were made in the description of the particle dynamics. In particular, the particle sinking velocity was described in the Stokes approximation. The aggregation and fragmentation of particles, mineral ballasting, ocean density stratification, and temporal changes in particle flows were not considered. While some simplifications can be eliminated by using a numerical approach, others require significant generalization. This applies particularly to the description of particle ballasting mechanisms. On the one hand, ballast affects the sinking of particles, but on the other hand, ballast minerals can protect organic matter from degradation (Cram et al., 2018). The processes of fragmentation and consumption of sinking particles, which are important in the upper mesopelagic layer, are poorly understood (Burd et al., 2024). Comparison of calculation results for different parameter values (e.g. $\eta$) did not reveal the advantage of one parameter value for both $C_p$ and $F_p$, which may be due to the incompleteness of the description of the processes of the simplified model used. Therefore, for the effective application of the proposed approach in biogeochemical models, a parameterization of the main process controls of the biological pump mechanism based on data from natural and laboratory measurements is necessary."

*Please include significance and follow-up work recommendations before the conclusion. The authors have shown appreciations of the limitations of the work at various places in the text, I'd recommend synthesizing them at the end before conclusions.*

**Answer.** Thank you for the suggestion. The reworked text is given in answer to the previous comment.

*In light of my comments above, I'd recommend minor revisions to the manuscript before it is published. Good luck and congratulations on this useful work.*

**Answer.** Thank you for the encouraging comments and suggestions.

---

## Author Response (AR1)

**Response to Reviewer #1**

We thank the reviewer for the careful reading and the suggestions to improve the quality and readability of the manuscript. We have followed your suggestions and revised the manuscript accordingly. Please find our responses below.

This is an interesting paper, examining a model of the vertical POC flux in the ocean below the euphotic zone. I found the paper hard to follow in places, and this was in part because of the choice of English usage: I would strongly urge the authors to seek out a native English speaker to clean this up.

**Answer.** The English usage was checked and improved with the help of a native English speaker.

There have been quite a few papers published recently that take a similar approach to the problem of modeling particle flux in the ocean, and the authors cite all of these (Kriest and Oschlies, 2008; Omand et al., 2020; DeVries et al., 2014). However, it is unclear what this manuscript presents that is new when compared with these other papers. Indeed, as far as I can see, there is no detailed comparison of results (except to show that one of their analytical solution is equivalent to that of DeVries et al.). I would like to see an analysis of what new things we learn from this model.

**Answer.** Thank you for your comment. Indeed, we have emphasised (L. 50) that we rely on the known parameterisations in the models (Kriest and Oshlies, 2008; DeVries et al., 2014; Cram et al., 2018). The novelty of our study is the development of the Euler-Lagrangian approach and the application of the corresponding numerical algorithm to solve the problem. We have added explanatory text.

L. 203 "Unlike the models (Kriest and Oshlies, 2008; Cael and Bisson, 2018) that use the same "Martin curve" power-law dependence (32) for the concentration and mass flux of POM with the exponent  $\beta$ , the exponent in the obtained solution (28) depends not only on  $\beta$  but also on the parameters that characterize the sinking velocity ( $\eta$ ) and the particle mass fractal dimension ( $\zeta$ )."

L. 345 "In this work, we considered a simple Eulerian-Lagrangian approach for solving equations that describe the gravitational sinking of organic particles under the effects of the sizes and ages of the particles, temperature and oxygen concentration on their dynamics and degradation processes. In contrast to other approaches, our approach does not use particle spectrum equations (e.g., DeVries et al., 2014) explicitly or power-law particle size distribution assumptions (e.g., Kriest and Evans, 1999; Maerz et al., 2020). Unlike (Omand et al., 2020), we do not assume a priori the constancy of the particle flux in depth in steady state problem. Instead, solutions are found for the Euler equation for the concentration of particles of a given size and the Lagrange equations for a sinking organic particle under the influence of microbiological degradation. In the stationary case, the problem is reduced to solving a system of ordinary differential equations of the first order, in contrast to (DeVries et al., 2014), where the solution of the hyperbolic equation of the first order for the particle distribution is found. In addition, the total concentration and flux of the POM are found by summation over the particle distribution at z' = 0, whereas in (DeVries et al., 2014) the summation is carried out over all depths. Our approach makes the particle transport model compatible with large-scale biogeochemical models and provides an opportunity to solve the non-stationary problem in the future using Eq. (1) complemented by the time derivative of  $C_{p,d}$  and necessary parameterizations of the POM sinking processes."

- L. 358 "Novel analytical solutions of the system of the one-dimensional Eulerian equation for the POM concentration and Lagrangian equations for the particle mass and depth were obtained for constant and age-dependent degradation rates..."
- L. 374 "A new Eulerian—Lagrangian numerical approach for solving the problem in general cases was presented. The algorithm includes time steps for Lagrangian variables (sinking velocity and particle mass) and Eulerian depth steps for the concentration of particles of size d. This enables the inclusion of different parameterizations of interacting degradation and sinking processes (e.g., DeVries et al., 2014; Cram et al., 2018; Omand et al., 2020; Alcolombri et al., 2021). However, in this study, we limited ourselves to the case where the degradation rate depends on the age of the organic particle, the temperature of the sea water and the concentration of oxygen. Notably, the developed numerical algorithm is suitable for arbitrary dependencies of mass and sinking velocity on the particle diameter. The proposed numerical method was tested on the obtained analytical solutions."

The model contains many assumption (as stated by the authors), but there is little to no analysis of the consequences of these assumptions. For example, everything is assumed to be a power-law (the mass-size relationship, the sinking velocity etc.) and while this makes things analytically tractable, it is unclear what observational evidence there is for them. For example, size distributions are often assumed to be power-law, but in reality this assumption often holds over a relatively small size range.

**Answer.** Thank you for your important comment.

Power dependencies are used for two reasons. The first reason is that the power law can be an effective method of parameterization, which, as the reviewer noted, allows one to obtain analytical solutions. Table 1 contains references to works that provide experimental data for the parameters of power dependencies. Note that the developed numerical algorithm suits arbitrary dependencies on the particle diameter. The second reason is that power dependences reflect fundamental properties of processes in nature, e.g. self-similarity of the formation of aggregates. The text and references to the papers with a critical analysis of these approximations were also added:

- L. 92 "The measurements of (McDonnell and Buesseler, 2010) show that formulations of sinking velocity as a function of only equivalent particle size can be insufficient because the shapes of the particles (e.g., faecal pellets) can significantly affect the sinking velocity. Fig. 1 from (Cael et al., 2021) also demonstrates the difficulties of describing the sinking velocities of particles of various sizes, shapes and 90 structures with a single universal dependence. Therefore, Eq. (4) should be considered only a first approximation when describing the complex dynamics of particles."
- L. 379 "Notably, the developed numerical algorithm is suitable for arbitrary dependencies of mass and sinking velocity on the particle diameter."

The model is a steady state model, and it's unclear if such an assumption is a reasonable one. For example, export fluxes out of the euphotic zone can vary significantly over time periods of days. So whilst I'm not opposed to the use of the steady state assumption, I do wonder about its validity.

**Answer.** Thank you for pointing out this issue. Yes, time-dependent export fluxes in bloom periods can be important factors in the euphotic layer and upper twilight zone in Polar oceans. Our approach to solving the model equations needs extension for non-stationary Eq. (1), which is out of this paper's scope. We added the text:

- L. 67 "We limit ourselves to large-scale climatological processes that cover the water column below the euphotic layer to the bottom. We assume that the effects of time variability on the POM flux are relatively small far from this layer, and we consider the steady states of these fluxes."
- L. 355 "Our approach makes the particle transport model compatible with large-scale biogeochemical models and provides an opportunity to solve the non-stationary problem in the future using Eq. (1) complemented by the time derivative of \$C\_{p,d}\$ and necessary parameterizations of the POM sinking processes."

Line 93, the mass loss is proportional to particle mass, not volume. The relationship in Equation (4) makes the correspondence between mass and volume unclear. For example, is the diameter the equivalent spherical diameter, is the volume the conserved volume or the encased volume?

**Answer.** We have made the following changes to the manuscript according to your comments:

- L. 96 "Parameter  $\theta = 1$  when the degradation rate is proportional to the particle mass, and  $\theta = 2/3$  when the degradation rate is proportional to the surface area of the particle (Omand et al., 2020)."
- L. 78 "The relationship between the organic matter mass *md* and diameter *d* of porous particles can be parameterized according to the particle fractal dimension."
- L. 70 "The Euler equation for the POM concentration  $C_{p,d}$  [kg m-3] for particles of equivalent spherical diameter d [m] is written as..."
- L. 105 "Furthermore, we suppose that the mass loss is proportional to the mass of the particle  $(\gamma = \gamma_0, \theta = 1)$  and does not depend on temperature or oxygen concentration  $(\gamma_0 = \text{const})$ ."

Line 109: I must be missing something here, because it's unclear to me that, practically, z-prime can never be larger than the inverse of psi. This follows from re-writing equation (10) and realizing that the constants eta, gamma0, and zeta are all positive. What am I missing?

**Answer.** For  $d_0 = 20 \cdot 10^{-6}$  m and  $d_0 = 200 \cdot 10^{-6}$  m, and for parameters  $\eta, \zeta, \gamma_0, c_w$  presented in Table 1 the values  $\psi^{-1}$  are 45.4 m and 672 m, respectively. Below these depths  $(z' > \psi^{-1})$ , only the trivial zero solutions for  $d, W_{p,d}, C_{p,d}$  has physical meaning. We added estimates for the layer thickness:

L. 192 "The finite thickness of the layer of sinking particles with parameters given in Table 1 varies in the range from 45.4 m at  $d_0 = 20 \,\mu\text{m}$  to 9937 m at  $d_0 = 2000 \,\mu\text{m}$ ."

The authors also need to make their notation more consistent. For example, in equation (15) we get the definition for  $C_{p,d}$ . But in equation (16) this becomes  $C_{p,d,i}$ . Also, in equation (16), n d becomes n. In equation (17) we are apparently integrating with respect to

a constant  $(d_0)$  having been defined as the initial particle diameter in equation (8)). So, the notation needs to be tidied up throughout the paper, not just in these places.

Answer. We corrected the notations accordingly to your comments.

z'=0. To obtain the size distribution of  $C_{p,d}(0)$ , we use a small increment of particle size  $\Delta d_0$  under the assumption that the concentration is uniform within the interval  $\Delta d_0$ . Then, the distribution  $C_{p,d}(0)$  is given by

$$C_{p,d}(0) = M_0 d_0^{-\epsilon} m_{0,d} \Delta d_0 = M_0 c_m d_0^{\xi - \epsilon} \Delta d_0.$$
(15)

The total concentration  $C_p$  is calculated as the sum of concentrations  $C_{p,k}$  in the k-th interval of size d over the total number of  $n_d$  intervals:

$$C_p(z') = \sum_{k=0}^{n_d} C_{p,k} = M_0 c_m \sum_{k=0}^{n_d} d_{0,k}^{\zeta - \epsilon} H(z') \left(1 - \psi z'\right)^{\frac{\zeta - \eta}{\eta}} \Delta d_0, \tag{16}$$

where  $d_{0,k} = k\Delta d_0 + d_0^{min}$ ,  $\Delta d_0 = (d_0^{max} - d_0^{min})/n_d$ , and  $d_0^{min}$  and  $d_0^{max}$  are the minimal and maximal values, respectively, of  $d_0$ . At  $\Delta d_0 \rightarrow 0$ , the total concentration of sinking POM  $C_p$  [kg m-3] in the range from  $d_0^{min}$  to  $d_0^{max}$  can be calculated as

$$C_p(z') = M_0 c_m \int_{d_0^{min}}^{d_0^{max}} \tilde{d}_0^{\xi - \epsilon} H(z') (1 - \psi z')^{\frac{\zeta - \eta}{\eta}} d\tilde{d}_0.$$
(17)

---

## Author Response (AR2)

**Response to Editor**

We thank the Editor for the careful reading and the suggestions to improve the quality and readability of the manuscript. We have followed your suggestions and revised the manuscript accordingly. Please find our responses below.

Dear authors,

After receiving one follow up review and reviewing the article myself, I am willing to accept the article for publication but only on the basis that the following revisions be completed:

1. Please describe more fully the physical meaning of  $\psi$  and  $\phi$  indices. Can you explain them intuitively to the reader.

Answer. Thank you for the suggestion. The text was added accordingly

- L. 125 "Solutions (11) and (12) describe the vertical distribution of  $W_{p,d}$  and d in the layer of finite thickness  $h_0 = \psi^{-1}$  below which there are only trivial solutions  $W_{p,d} = 0$  and d = 0."
- L. 167 "The parameter  $\phi$  [m-1] characterizes the vertical scale of attenuation  $W_{p,d}(z')$ ,  $\gamma(z')$  and d(z') with depth."
- 2. Please explain more intuitively in words what is happening in Eq. 9, Eq. 13, Eq. 15, Eq. 27, so that a less quantitative reader can follow.

**Answer.** We reworked the text for these equations according to your suggestions:

Assuming the quasiequilibrium sinking of the particle in the Stokes regime, as described by Eq. (4), and taking into account that particle trajectory in Lagrangian system of coordinates is described as  $\partial z'/\partial t = W_{p,d}$ , we estimate the dependence of the particle depth z' on t using Eqs. (4) and (8):

$$\frac{\partial z'}{\partial t} = c_w d_0^{\eta} \exp\left(-\frac{\eta \gamma_0 t}{\zeta}\right). \tag{9}$$

By integrating Eq. (9) from the initial particle depth z' = 0 at t = 0, we find the vertical path travelled by the particle:

$$z' = \frac{\zeta c_w d_0^{\eta}}{\eta \gamma_0} \left[ 1 - \exp\left( -\frac{\eta \gamma_0 t}{\zeta} \right) \right]. \tag{10}$$

By eliminating time from Eqs. (8) by using Eq. (10) and then substituting Eq. (8) to Eq. (4), we obtain  $W_{p,d}$  and d as functions 120 of z':

- L. 122 See the reply to the previous comment.
- L. 135 "Then, the distribution  $C_{p,d}(0)$  can be represented as a product of particle size distribution  $N(d_0)$  and mass of particle  $m_{0,d}$  ...."
- L. 166 See the reply to the previous comment.
- 3. Please label the equation on line 128 and explain what your assumptions are: i.e., that particle number decreases with particle size according to some power law scaling.

**Answer.** Thank you for the suggestion. The text was clarified accordingly:

- L. 131 "The distribution  $N(d_0)$  was approximated in such a way that the number of particles decreased with increasing particle size according to power law scaling."
- 4. What is AIDR and ADDR? You must explain what these are.

**Answer.** We have corrected both AIDR and ADDR to AID and ADD.

5. Please add the results from your sensitivity analysis to the main paper as a figure.

**Answer.** We added the results of the sensitivity analysis in the paper as Fig. 7.

6. Line 348 - please be clear that you also make assumptions about the size spectrum obeying a power law.

**Answer.** We have clarified the text accordingly:

L. 353 "In contrast to other approaches, our approach does not solve particle size spectrum equations (e.g., DeVries et al., 2014) explicitly or introduce power-law particle size distribution assumptions below the euphotic layer (e.g., Kriest and Evans, 1999; Maerz et al., 2020). Note that the particular form of size spectrum dependence  $N(d_0)$  may differ from the power law (15)."

7. Line 356 - Please give a more concrete example of how this formulation would be advantageous to ocean biogeochemical models. What would implementing your approach allow? What questions might it provide answers to?

**Answer.** We added text accordingly.

L. 362 "As shown in Table 3 from the review (Burd, 2024), the sinking velocity in CMIP6 Eulerian biogeochemical models is either assumed to be constant or it increases linearly with depth. Our hybrid approach considers the interaction between the sinking and degradation processes of POM particles in Lagrangian variables and POM concentration in the Eulerian coordinate system, making particle transport models compatible with large-scale Eulerian biogeochemical models. It also provides an opportunity to solve the non-stationary problem in the future using Eq. (1) complemented by the time derivative of  $C_{p,d}$  and necessary parameterizations of the POM sinking processes."